# Facial emotion recognition abilities of individuals with schizophrenia and the influence of parental bonding—An exploratory study in a forensic sample

**Salome Todua-Lennigk**[1,2]*, **Gunnar Deuring**[3], **Marc Graf**[1], **Henning Hachtel**[2]

**1** University of Basel, Basel, Switzerland, **2** Forensic Department, University Psychiatric Clinic Basel, Basel, Switzerland, **3** Center of Affective, Stress and Sleep Disorders (ZASS), University Psychiatric Clinics (UPK), University of Basel, Basel, Switzerland

* salome.todua-lennigk@upk.ch

## Abstract

Individuals diagnosed with schizophrenia experience cognitive impairments, including a decline in social cognition, which encompasses facial emotion recognition. Facial emotion recognition is an important aspect of social interaction, guiding people's actions and influencing their social functioning. Early childhood experiences, such as parental attachment, are one of the most influential factors in the development of many psychiatric symptoms including impairment of social cognition. Our aim was to explore this poorly researched area. We investigate the hypothesis that dysfunctional parenting styles negatively affect facial emotion recognition abilities in general and further worsen these deficits in individuals diagnosed with schizophrenia. A total of 32 participants were included in an exploratory study, comprising 16 patients with paranoid schizophrenia recruited from forensic clinics in Switzerland, and 16 age and education matched healthy controls without history of psychiatric or neurological illnesses. Parental attachment was assessed using the Parental Bonding Inventory and subjects were assigned to subgroups of optimal vs. neglectful parenting style from both parental sides. Facial emotion recognition was operationalized as the error rate in an emotion-naming task using standardized images of the five basic emotions. Overall, schizophrenia patients made significantly more errors in the facial emotion recognition task than healthy controls. Interestingly, in the subgroup with optimal parental attachment experiences, patients did not significantly differ from controls, whereas in cases of neglectful parenting, the patients showed a much higher error rate in facial emotion recognition compared to healthy controls ($p < .001$), as well as compared to patients with optimal parenting experience ($p < .01$). Neglectful parenting appears to exacerbate the adverse effects of schizophrenia on facial emotion recognition; i.e., optimal parenting might mitigate deficits caused by schizophrenia spectrum disorder patients and help compensate for FER impairments.

**Data availability statement:** All relevant data are within the manuscript and its Supporting information files.

**Funding:** This work was supported by the Research Promotion Fund of the Psychiatric Clinics of the University of Basel. The authors declare that they have received financial support for the research.

**Competing interests:** The authors have declared that no conflicts of interest exist.

## Introduction

Schizophrenia, which is the one of most common types of schizophrenia-spectrum disorder, is a chronic psychiatric illness [1]. It has a prevalence rate of 1% and is one of the most severe and disabling diseases in psychiatry [1,2]. Paranoid schizophrenia is one the subtypes of schizophrenia and play by far the largest role in everyday clinical practice [3]. It is characterized by the occurrence of so-called positive symptoms (e.g., hallucinatory symptoms, delusional symptoms) and negative symptoms (e.g., avolition, affective flattening) [1,2].

Patients with schizophrenia experience cognitive impairments, including impairments in social cognition, which encompasses the mental processes required to perceive, interpret, and process information for successful social interactions, e.g., emotion processing and facial emotion recognition [FER], theory of mind, attribution bias, and social perception [4–6].

Several studies have suggested that patients with schizophrenia-spectrum disorder experience impairment in cognitive functioning, including the domains of higher cognitive functions and social cognition [7–14]. Impairment in social cognition occurs in the early stages as well as throughout the course of the disorder [8–11]. The current evidence on whether social cognitive impairment worsens during the disease course is heterogeneous: some studies show that cognitive functional impairment remains stable, whereas others have concluded that patients with chronic schizophrenia tend to have poorer cognitive functioning than those experiencing their first episode [15,16].

Recognizing emotions in interaction partners is one of the central processes in social interaction and one of the most frequently investigated aspects of social cognition in schizophrenia [4,17–19]. Emotion perception and recognition of facial expressions is a key component of both social cognition and therefore nonverbal communication, which helps us to understand the affective and mental states of others [4,6,17,18] and respond appropriately [17,19]. Thus, emotion recognition is associated with social behaviour, social functioning, and, consequently, social integration and well-being [4,5,20,21].

FER is providing essential information for the regulation of social interactions [22]. Several studies have further demonstrated that individuals with schizophrenia exhibit deficits in FER that correlate with deficits in executive function [11,23,24]. Similarly to social cognition in general, FER impairment is manifest in individuals with schizophrenia before the onset of psychotic episodes and was found to remains stable [25], while other studies have shown that the severity of FER deficits appears to increase with the number of psychotic episodes experienced [9,26].

In patients with schizophrenia, FER impairment is correlated with negative symptomatology and disorganization. However, these correlations exhibit small effect sizes [11,27]. FER deficits are also associated with various psychiatric disorders, socialization difficulties, and violent tendencies in schizophrenic patients [28,29].

An individual's early childhood experiences, specifically parental attachment, are another- important factor that substantially impacts social cognition [30–36] and the

development of various psychiatric symptoms and disorders [37–45]. Bowlby first described the different attachment patterns and their role in the development of psychiatric disorders [46–49]. His observations were later supported by various studies, notably those on affective disorders [37,40], certain anxiety disorders [37], personality disorders, such as borderline personality disorder [38], and eating disorders in women [39]. Various studies suggest that neglectful care, particularly by mothers, has adverse effects on the lifetime prevalence of affective disorders, such as depression or anxiety, in a large population across Western European countries where there is no significant cultural dissimilarity [41–44]. Furthermore, associations have been found between neglectful parenting, childhood trauma, and the independent development of psychotic symptoms, regardless of the underlying diagnosis [41–44].

Individuals with schizophrenia-spectrum disorder, including schizophrenia often report dysfunctional parental attachment or neglect. However, current information on the detrimental effects of parental attachment on later social cognition, including FER, tends to be limited [30–36].

Based on the findings outlined above and on the knowledge that this area has rarely been the subject of study and that there is limited understanding of the factors influencing FER, we have decided to explore this area. To work towards this aim, we have formulated the following hypotheses regarding FER in patients with schizophrenia and the effects of early childhood experiences, such as parental attachment style:

a. Dysfunctional parenting styles generally worsen FER.

b. Dysfunctional parenting may worsen the negative impact of schizophrenia on the ability to recognize facial emotions.

## Materials and methods

### Design, participants, and procedure

This paper is a secondary, exploratory analysis of data from a study, which was itself conducted for the purpose of generating new insights into the subject matter. Women are a minority in forensic settings [50] and to control for the potential confounding effects of sex on facial emotion recognition [51,52], a male-only sample was planned for our cross-sectional case-control design. Sixteen male inpatients with a diagnosis of paranoid schizophrenia (PAT) were recruited from the forensic psychiatric clinics of Basel and Königsfelden (Switzerland) – forming a forensic convenience sample. We chose individuals with paranoid schizophrenia for the study because they play by far the largest role in everyday clinical practice [3] and this is the predominant type of schizophrenia [53]. The Exclusion criteria for PAT – Group was: Potential patient participants were excluded based on a history of psychotic disorders because of organic pathology, substance abuse, affective disorders, or emotionally unstable personality disorders of the borderline type or if they exhibited mental disability, neurological disorders, current benzodiazepine use, or poor comprehension of the German language. A forensic study population in contrast to the population in general psychiatry, is characterized by comprehensive evaluation and a reliable diagnosis through systematic assessments, tight and detailed documentation of the course of the illness, including clinical symptoms and medication or drug intake. This ensures that the study population is closely monitored, thereby minimizing the risk of medication or substance misuse. The mean illness duration were 11.6 years (SD 7.9 years), the daily dose equivalent of antipsychotic medication was chlorpromazine 618 mg (SD = 438 mg) [54]. The mean duration of antipsychotic intake were 10 years (IQR 10,8 Years) (see Table 1). To date, no studies have confirmed a negative effect of antipsychotic medication on facial emotion recognition. Studies show that antipsychotic treatment has little positive effect on facial emotion recognition skills, so no definitive conclusions can be drawn about their potential influence on facial emotion recognition [54]. Therefore, to our current knowledge no drug-related bias in the results can be assumed. A healthy control group (CTL) was recruited through advertisements; participants in this group were also male and were matched to the patient group by age and years of formal education. The exclusion criteria for the control group were schizophrenia spectrum disorders, index diagnoses of affective disorders, substance abuse, mental disability, personality disorders, a history

**Table 1. Demographics and sample characteristics by group.**

| | Patients N = 16 | | Controls N = 16 | | | |
|---|---|---|---|---|---|---|
| | Mean | SD | Mean | SD | p* | IQR |
| Age (years) | 33.2 | 8.0 | 32.8 | 10.6 | .91 | |
| Education (years) | 11.7 | 2.5 | 12.5 | 2.1 | .34 | |
| Illness duration (years) | 11.6 | 7.9 | | | | |
| PBI mother care | 23.4 | 6.8 | 24.1 | 8.3 | .60 | |
| PBI mother overprotection | 11.4 | 5.0 | 11.1 | 7.8 | .36 | |
| PBI father care | 22.9[1] | 7.0 | 21.7 | 7.0 | .65 | |
| PBI father overprotection | 10.4[1] | 5.2 | 8.3 | 3.7 | .21 | |
| PANSS-positive scale | 12 | 4.0 | | | | |
| PANNS-negative scale | 15.9 | 5.2 | | | | |
| BPRS | 31.5 | 4.5 | | | | |
| Chlorpromazine equivalent (mg) | 618 | 438 | | | | |
| Medication intake (years) | 10 | | | | | 10.08 |

[1] N = 14, 2 × no report on paternal bonding style; *Student's t-test or Wilcoxon rank-sum test was applied for group comparisons depending on the data distribution.

of traumatic brain injury, neurological disorders, current benzodiazepine use, and poor comprehension of the German language.

The data presented here were collected as a secondary endpoint in the study on the impact of psychosocial stress on facial emotion recognition in schizophrenia by Hachtel et al. [55]. The study was approved by the Ethics Committee of Northwestern and Central Switzerland. All participants were included between 7 September 2016 and 24 May 2017. They were informed about the study procedures and provided written informed consent. The capacity to consent was assessed and confirmed by the treating physicians for participating patients. The authors had access to information that could identify individual participants during or after data collection.

## Methods and measures

The participants' demographics characteristics and the retrospective measure of Parental Bonding Inventory (PBI) were recorded on a first session, followed by a separate assessment of FER and some other tasks that are not the subject of this paper and are described by Hachtel et al. [55].

The clinical symptoms and psychopathology were evaluated using two scales: the Brief Psychiatric Rating Scale (BPRS) [56] and the Positive and Negative Syndrome Scale (PANSS) [57]. The level of psychopathology in the PAT group was low (see Table 1).

The Parental Bonding Inventory (PBI) is a self-report questionnaire to assess parental care and overprotection [58]. The PBI aims to retrospectively quantify perceived parenting styles during infancy and adolescence. The instrument consists of two parts; each composed of 25 items, concerning the respondent's perception of their mother's and father's parenting styles. The perceived parenting style is measured on the two dimensions of care and overprotection. The care dimension concerns the sense of warmth and perceived emotional involvement, whereas overprotection explores perceived parental control. The scores for each dimension are categorized according to original PBI publication [58] as low or high, and the cut-off scores for the maternal and paternal sides differ. For paternal bonding, the cut-off scores are < 24.0 for low care, ≥ 24.0 for high care, < 12.5 for low overprotection, and ≥ 12.5 for high overprotection. For the maternal side, the cut-off scores are < 27.0 for low care, ≥ 27 for high care, < 13.5 for low overprotection, and ≥ 13.5 for high

overprotection. Four parenting styles were obtained by crossing these two dimensions: *optimal parenting* (high care, low overprotection), *neglectful parenting* (low care, low overprotection), *affectionate constraint* (high care, high overprotection), and *affectionless control* (low care, high overprotection) [58].

To evaluate the recognition of emotions from facial expressions, we employed specific FER tasks based on subtest A from Comparelli et al. [25]. Subtest A is a verbal naming task that consists of 80 images from Karolinska's standard image set [59]. The facial expressions of the five base emotions (anger, happiness, sadness, fear, disgust) were according to the specified procedure presented on the computer screen seven times in random order for subtest A. The subjects had to press the key that was associated with the emotion within the time limit of 2,000 ms, after which any response was considered incorrect. The task consisted of 35 trials (7 × 5) with a possible score range of 0–35 (see Fig 1).

## Statistical analysis

The statistical analysis was conducted with R version 4.0. Univariate group comparisons were performed using Student's t-test for independent samples with normally distributed data and the nonparametric Wilcoxon rank-sum test otherwise. The ability to recognize facial emotions was operationalized as the error rate in the FER facial emotions naming task (FERerr). The effect of *parenting style* in the two *groups* on FERerr was analyzed. The FERerr distribution was right-skewed. A Box–Cox distribution analysis resulted in a lambda of 0.37, which is close enough to 0.5 to approximate the error distribution of a generalized linear model (GLM) with the Gaussian family square root link function ("sqrt"). The GLM main and interaction effects were tested by a type-3 analysis of deviance (AoD) using the *Anova* function from the *car* package [60], and a simple main effects post hoc contrast analysis was conducted using the package *emmeans* [61]. Cohen's partial omega squared ($\omega_p^2$) is reported as effect sizes for the GLM using the package *effectsize* [62]; $\omega_p^2$ is interpreted as follows, comparable to eta squared: small effect, $\omega_p^2 \geq .01$; medium effect, $\omega_p^2 \geq .06$; large effect, $\omega_p^2 \geq .14$ [63]. For post hoc contrasts, effect sizes are reported as Hedges' *g* (small effect: $g \geq .20$, medium effect: $g \geq .50$, large effect: $g \geq .80$, comparable to Cohen's *d* [63]. All tests were two-sided, and *p*-values below.05 were considered statistically significant.

## Results

The 16 patients and 16 control subjects were matched by age and education (see group statistics in Table 1).

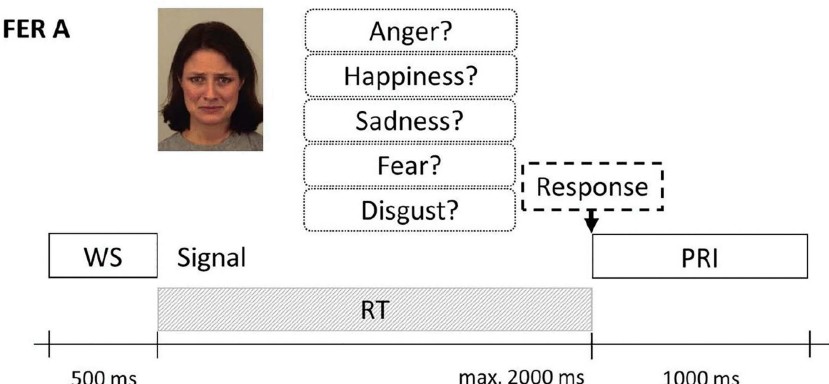

**Fig 1. Example of a verbal recognition task.** The signal and response options are shown simultaneously until a response is selected or the time limit is reached. WS: warning signal, RT: reaction time, PRI: post-response interval. Stimulus images were taken from the Karolinska Directed Emotional Faces set (Lundqvist, 1988); ID here: AF01SAS.

Spearman's ρ correlation coefficients between the PBI scores and FERerr, for both groups pooled, and each group separate, ranged between absolute 0.07 and 0.36, while none of them was statistically significant ($0.72 \geq p \geq 0.17$; see S4 Table).

The PBI parenting style *affectionate constraint* did not occur in the sample. Two patients did not report on the paternal side, so the two missing data points were filled in using the maternal parenting style responses. Table 2 shows the frequencies of parenting styles in both groups. The parenting style *affectionless control* occurred only four times in the control group. As this imbalance severely impairs our ability to conduct a meaningful analysis, this factor level was omitted from the analysis. Only the optimal attachment style and the neglectful attachment style were finally included.

The AoD main effects of the GLM *group* and *parenting style* as well as the interaction effect were significant (*group*: $F_{(1,46)} = 19.02$, $p < .0001$, $\omega_p^2 = 0.28$, large effect; *parenting style*: $F_{(1,46)} = 4.07$, $p = 0.049$, $\omega_p^2 = 0.06$, medium effect; *group × parenting style*: $F_{(1,46)}$, $p = .03$, $\omega_p^2 = 0.1$, medium effect; see S1 Table). The post hoc comparisons indicated that the estimated marginal means of FERerr were almost the same for CTL under both *parenting style* conditions ($p = .87$). In the optimal parenting condition, PAT values were elevated but not significantly higher than those of CTL ($p = .13$), whereas under the neglectful parenting condition, the PAT FERerr was significantly higher than that of CTL ($t_{(46)} = 4.66$, $p < .001$, $g = 1.96$, large effect) and significantly higher than PAT under optimal parenting conditions ($t_{(46)} = 3.267$, $p = .001$, $g = 1.34$, large effect; see Fig 2 and S2 and S3 Tables). The GLM assumptions were evaluated and found well met, and the Nagelkerke $R^2$ was 0.41.

Parent specific effects were not found in our sample, a GLM including *parent* (father/mother) as additional factor did not show a significant effect for *parent*, ($F_{(1,45)} = 0.099$, $p = 0.75$), while the main effect *group* and the interaction effect *group × parenting style* remained unchanged; the marginally significant main effect *parenting style* was not significant anymore, $p = 0.06$, due to model complexity.

## Discussion

Our study aimed to investigate the association between parental bonding, schizophrenia, and FER. One result of the current investigation is that under optimal parental attachment conditions, the patients with schizophrenia did not show significantly higher FERerr compared with the healthy controls. However, the present study did not support our hypothesis that non-optimal parental attachment generally affects facial emotion recognition abilities. One explanation for the finding could be that patients learn effectively from parental role models under conditions of optimal parental attachment before the disease onset. However, patients who experienced neglectful parenting did have a significantly higher FERerr compared with controls. A possible explanation for this finding could be that patients affected by poor parental care, unlike healthy individuals, are unable to compensate for their learning deficits through social contacts other than their parents. This finding is consistent with the results of previous studies, such as Rokita et al. [30,64], which also investigated the direct association between parental bonding and adult FER, including among patients with schizophrenia, and reached the same conclusion: the effect of childhood trauma on emotion recognition was attenuated by optimal parental bonding

**Table 2. Parenting styles by group, maternal and paternal scores combined.**

| Parenting style | Group | | $\chi^2_{(1)}$ § | P |
| --- | --- | --- | --- | --- |
| | $N_{CTL}$ | $N_{PAT}$ | | |
| *Optimal parenting* | 18 | 11 | 1.69 | .19 |
| *Affectionless control* | 4 | 10 | 2.571 | .11 |
| *Neglectful parenting* | 10 | 11 | 0.048 | .83 |

§Chi-squared test for given probabilities of 0.5 vs. 0.5; *df* = 1.

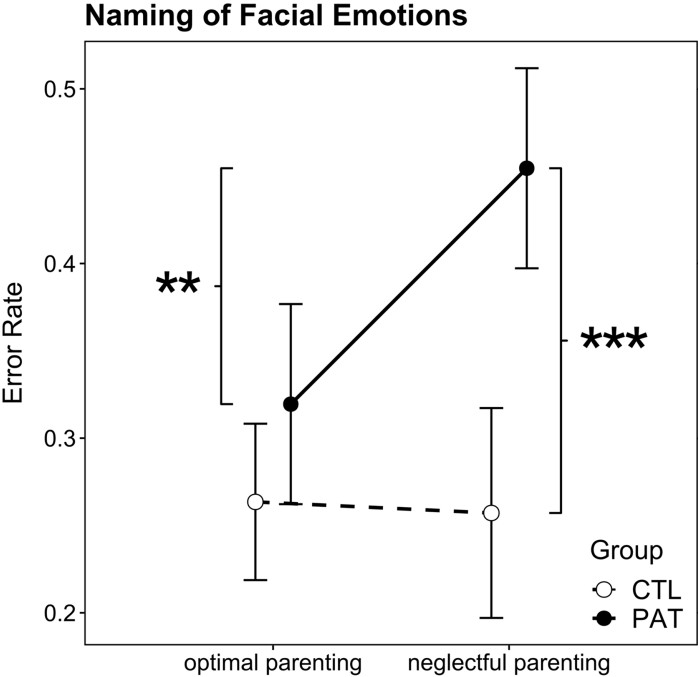

**Fig 2. Estimated marginal means of the facial emotion recognition error rate regarding the naming of facial emotions task in the control (CTL) and patient (PAT) groups for the conditions *optimal parenting* and *neglectful parenting*.** The vertical bars denote 95% confidence intervals, and significance levels are indicated as ** $p < .01$, *** $p < .001$.

[64]. Although they did not directly investigate FER and parental attachment, the authors found an association between non-optimal parental bonding, as well as traumatic childhood experiences and neglect and impaired social cognition including theory of mind, of which FER is a subset [4,6,17,18].

In contrast to the aforementioned studies, Palmier-Claus et al. [35] and Kilian et al. [33] found no significant association between theory of mind or social cognition and childhood trauma.

However, apart from the study by Rokita et al. [64], no other investigators [32–36] have explored the direct association between parental attachment and FER. Instead of parental bonding specifically, they focused on the association between childhood experiences, such as trauma or neglect, and social cognition [32–36]. These previous studies also did not specifically examine emotion recognition. Our study sought to fill this gap and focused instead on the direct association between parental attachment and FER, as did the study by Rokita et al [64], while investigating a forensic sample. However, it should be emphasised that this study did not investigate a causal relationship between parental attachment and the ability to recognise facial emotions, merely an association.

## Strengths and limitations

The strength of our study is its case-control design; controls were carefully selected to match for age and years of formal education. The tests used are validated. A forensic sample contrasts with the population in general psychiatry characterized by reliable diagnostics and monitoring of drug abstinence or medication intake, thereby minimising the risk of medication or substance misuses.

The study has some limitations, including its small sample size, single-gender group (male), convenience sampling and cross-sectional design, which do not examine causality, but only statistical correlations. Due to the small sample size, one

parenting style category (*affectionate constraint*) was not observed and one (*affectionate constraint*) had too few observations in the control group to allow for a reliable analysis. As we did not have a female group, it is difficult to generalise the results to the other genders. However, most studies have indicated that women tend to perform better at recognising facial emotions, characterised by enhanced accuracy and speed, across a range of emotions and expression intensities [65–67]. Nevertheless, certain studies have found that this advantage is negligible and not consistently present [68–70].

As we recruited individuals with schizophrenia from a forensic setting, the results are not generalizable to other samples of individuals with schizophrenia without consideration. Moreover, the PBI is a self-reported, retrospective test that may be susceptible to memory bias [71]. The PBI has been shown to demonstrate good internal consistency and construct validity in cases of psychosis and schizophrenia. It has been utilized as a tool to differentiate between clinical and non-clinical groups, with the distinction being based on the experiences of parental bonding reported by the participants [72–75]. However, PBI is an instrument that is widely used in research for the evaluation of parental attachment, and we are not aware of any more objective instruments for the assessment of parental attachment.

## Conclusion/Outlook

Future studies with larger and more diverse samples could provide a more nuanced understanding of the influences of various parenting styles on different genders and would facilitate future meta-analyses and help to verify the robustness of the current findings.

It is important to note that this study did not investigate causality, which could be a focus of future research. For instance, future studies might investigate potential structural brain alterations linked to impaired emotion recognition in individuals who report having experienced neglectful parenting.

The practical applications of our results may include the following:

- Using FER impairment as a diagnostic tool for early detection of schizophrenia, particularly in vulnerable populations, such as ultra-high risk (UHR) individuals, those with a family history of the disease, and younger siblings of patients. Empirical evidence shows deficits in FER early in the disease or before symptom onset [13], in this context the early detection of deficits in facial emotion recognition could be a helpful addition to the early detection of schizophrenia.

- The development of early therapy to enhance FER and social skills in individuals who are already ill (existing therapy approaches with neuromodulation demonstrated notable effects on recognition of facial emotions, see Yamada et al., [21]) may facilitate improved quality of life and, in a forensic context, contribute to the reduction of violent tendencies. According to Bulgari et al, deficits in facial emotions are associated with higher tendencies towards violence [28]; it can therefore be assumed that improving these deficits could lead to a reduction in violent tendencies.

- The social learning behaviour of individuals with schizophrenia undergoes significant changes following the onset of the illness, primarily due to impairments in social cognition, emotion recognition and theory of mind. While premorbid social learning is usually intact, deficits in interpreting social cues and adapting behaviour are common post-illness [76–80]. Supporting parenting of children of ill parents (UHR), in the sense of informed parenting. There is evidence that parental care is not optimal in UHR individuals [73], which leads to a higher error rate in FER in the case of schizophrenia, as described above. Informing parents about the possible effects of parental attachment or care and supporting them in the everyday upbringing of children with UHR could lead to an improvement in parental attachment. This in turn could lead to an improvement in the ability to recognise facial emotions.

- Improving family cohesion and strengthening the family system by educating family members of individuals with schizophrenia about FER deficits. The misinterpretation of facial emotions leads to interpersonal and socialisation problems [4,5,20,21], which could be a burden for the family. Sufficient education of family members would improve understanding of the problem and simplify the handling of it.

In summary, our findings provide further evidence of a link between parental bonding styles and FER in individuals with schizophrenia. The results do not indicate a general influence of dysfunctional or neglectful parenting on FER as hypothesized, except when combined with schizophrenia. Consequently, our main results demonstrate that patients with schizophrenia who suffered under neglectful parenting exhibited a higher rate of error in recognizing facial emotions. This outcome could mean that, unlike healthy individuals, patients affected by poor parental care may not be able to compensate for learning deficits through other social contacts besides their parents.

## Supporting information

**S1 Table. Analysis of Deviance Table for the GLM tested against the grand mean of FER an error rate.**
(DOCX)

**S2 Table. Estimated marginal means of FER error rate in both groups under both parenting styles.** CL: 95% confidence limits for means.
(DOCX)

**S3 Table. Simple main effect contrast estimates.**
(DOCX)

**S4 Table. Spearman's ρ correlation coefficients between the PBI scores and FERerr.**
(DOCX)

**S5 Table. A comparison of recent studies of facial recognition (FER) impairment in schizophrenia and control groups.**
(DOCX)

**S1 File. List of Karolinska Directed Emotional Faces images used in FER A.**
(DOCX)

**S2 File. Statistical Analysis Pipeline of the GLM in R (core components).**
(DOCX)

## Acknowledgments

The authors would like to thank Peter Wermuth and his staff in clinic Königsfelden for help in recruiting patients.

The authors acknowledge BioScience Writers, LLC, for contributions to technical editing of the manuscript.

## Author contributions

**Conceptualization:** Henning Hachtel.

**Data curation:** Salome Todua-Lennigk, Gunnar Deuring.

**Formal analysis:** Salome Todua-Lennigk, Gunnar Deuring.

**Investigation:** Salome Todua-Lennigk, Gunnar Deuring, Henning Hachtel.

**Methodology:** Gunnar Deuring, Henning Hachtel.

**Project administration:** Gunnar Deuring, Henning Hachtel.

**Resources:** Gunnar Deuring, Henning Hachtel.

**Software:** Gunnar Deuring.

**Supervision:** Henning Hachtel.

**Visualization:** Gunnar Deuring.

**Writing – original draft:** Salome Todua-Lennigk, Gunnar Deuring, Marc Graf, Henning Hachtel.

**Writing – review & editing:** Salome Todua-Lennigk, Gunnar Deuring, Marc Graf, Henning Hachtel.

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
