## [Decision Letter · Decision Letter 0]

22 Jul 2025

Dear Dr. Todua-Lennigk,

Thank you for submitting your manuscript to PLOS ONE. After careful consideration, we feel that it has merit but does not fully meet PLOS ONE’s publication criteria as it currently stands. Therefore, we invite you to submit a revised version of the manuscript that addresses the points raised during the review process.

We look forward to receiving your revised manuscript.

Kind regards,

Mu-Hong Chen, M.D., Ph.D.

Academic Editor

PLOS ONE

Journal Requirements:

2. Please describe in your methods section how capacity to provide consent was determined for the participants in this study. Please also state whether your ethics committee or IRB approved this consent procedure. If you did not assess capacity to consent please briefly outline why this was not necessary in this case.

[This work was supported by the Forschungsförderungsfonds of the University Psychiatric Clinics of Basel. The authors declare financial support was received for the research.].

4. We note that Figure 1 includes an image of a patient/participant in the study.

Reviewers' comments:

Reviewer's Responses to Questions

**Comments to the Author**

1. Is the manuscript technically sound, and do the data support the conclusions?

Reviewer #1: Yes

Reviewer #2: Yes

Reviewer #3: Yes

2. Has the statistical analysis been performed appropriately and rigorously?

Reviewer #1: Yes

Reviewer #2: Yes

Reviewer #3: Yes

3. Have the authors made all data underlying the findings in their manuscript fully available?

Reviewer #1: Yes

Reviewer #2: No

Reviewer #3: Yes

4. Is the manuscript presented in an intelligible fashion and written in standard English?

Reviewer #1: Yes

Reviewer #2: No

Reviewer #3: Yes

Reviewer #1: I have reviewed the manuscript titled “Facial Emotion Recognition Abilities of Individuals with Schizophrenia and the Influence of Parental Bonding – An Exploratory Study in a Forensic Sample.” The study addresses an important and underexplored area, investigating how parental bonding patterns influence facial emotion recognition (FER) performance in individuals with schizophrenia within a forensic sample. The findings are clinically relevant, as they suggest that optimal parental attachment may mitigate FER deficits, while neglectful parenting is associated with poorer performance.

To further strengthen the study, I suggest the authors consider sharing their full analysis code and data processing pipeline (where ethically possible) to promote reproducibility. Including a more detailed subgroup or ablation analysis could help disentangle the specific contributions of different parental bonding dimensions (care vs. overprotection) to FER outcomes. Validation on independent or more diverse cohorts would also help confirm the robustness of the findings. The discussion would benefit from a state of the art comparison table summarizing existing FER studies in schizophrenia to better contextualize the current results. A dedicated limitations and future work section should be expanded to explicitly address issues such as sample size, gender restriction, retrospective bias in PBI reporting, and generalizability. Finally, the inclusion and exclusion criteria for participant selection should be clearly summarized for clarity.

For further methodological inspiration, particularly on advanced feature extraction and classification in psychiatric populations, the authors may consider reviewing the following works: “Zipper Pattern: An Investigation into Psychotic Criminal Detection Using EEG Signals,” “Multilevel Hybrid Handcrafted Feature Extraction Based Depression Recognition Method Using Speech,” and “Automated Schizophrenia Detection Model Using Blood Sample Scattergram Images and Local Binary Pattern.” These studies illustrate how explainable and structured computational models can enhance the understanding and classification of complex psychiatric and neurocognitive phenomena offering valuable insights for future extensions of this important research on schizophrenia and emotion processing.

Reviewer #2: The article addresses a relatively unexplored issue: the role of parental attachment in facial emotion recognition (FER) among individuals with schizophrenia in forensic settings. Below some methodological and scientific limitations:

1. The sample size (n = 32) is limited and does not allow for generalizable conclusions. Additionally, one parenting style category (“affectionate constraint”) is not represented in the sample. Please, include a power analysis to justify the sample size or explicitly acknowledge the lack of such analysis as a limitation.

2. Although justified to minimize gender-related biases in FER, this choice significantly limits the generalizability of the findings. Please discuss in more depth the potential gender differences in schizophrenia spectrum disorders and in facial emotion recognition.

3. The use of the PBI, a retrospective self-report instrument, may introduce recall bias or social desirability distortions. to encompass these limits, please cite studies validating the PBI in schizophrenia populations and strengthen the discussion of its methodological limitations.

4. The study does not report on important variables such as current symptomatology (e.g., PANSS), Functioning level (e.g., GAF, PSP) and ongoing pharmacological treatment (type, dosage, duration); please explicitly acknowledge them as potential uncontrolled confounders.

5. Subtest A of the emotional recognition test includes only five basic emotions and a limited number of stimuli. It omits more complex emotions (e.g., contempt, surprise) and variations in intensity. You can richer data incorporating measures such as reaction times and confidence ratings.

6. Although the authors acknowledge the correlational nature of the study, certain statements (e.g., “optimal parenting could ameliorate...”) may be interpreted as causal or overly speculative. I think that it will be better to reformulate these statements with greater caution, using expressions such as “could be associated with...” or “might mitigate...”.

Additional suggestions for improving the scientific strenght of the paper would be:

1.to include correlations between PBI scores and FER error rates (FERerr) within each group, not just main effects and interactions;

2.to provide a more detailed discussion of the forensic context: what distinguishes these patients from non-forensic individuals with schizophrenia?

3.to examine potential differences in FER error patterns—for example, whether certain emotions such as fear are systematically misrecognized, as suggested by previous studies.

4.to discuss the role of childhood trauma separately from parental bonding to avoid conceptual confusion between the two constructs.

Overall the article addresses an original and relevant topic in forensic and social psychiatry and is methodologically well-structured. However, it presents several limitations related to sample size, retrospective assessment tools, generalizability, and causal interpretation. With some revisions, this study could make a significant contribution to the understanding of the role of early parental relationships in the social cognition of individuals with schizophrenia, particularly in high-complexity settings such as forensic contexts.

Reviewer #3: The study explores whether retrospectively assessed parenting style moderates facial emotion recognition (FER) deficits in men with paranoid schizophrenia recruited from Swiss forensic clinics. Sixteen patients and 16 age- and education-matched healthy males completed the Parental Bonding Instrument and a 35-trial FER. A general linear model showed significant main effects of group and parenting style, plus an interaction whereby neglectful parenting was linked to markedly higher FER error rates among patients but not controls. The authors conclude that optimal parental bonding may buffer FER impairment in schizophrenia and suggest clinical and forensic implications. The topic is novel, the paper is clearly organized, and the forensic sample adds ecological relevance for violence-risk research. Nonetheless, several methodological and interpretive issues warrant clarification, as detailed below.

- Prior FER literature is largely cited, but the authors focus exclusively on male forensic patients to control for the potential confounding effects of sex. Did the authors hypothesize that violent behavior would moderate the relationship between schizophrenia and facial emotion recognition, and if so, could they clarify how this informed the study design?

- Sample size (N = 32) is small; was an a priori power calculation performed? Multiple post-hoc contrasts were run—were p-values adjusted (e.g., Bonferroni or Holm) to control Type I error?

- The PBI’s 25-item version is specified, yet cut-offs are provided without citation; would the authors report reliability coefficients (Cronbach’s α) for care/overprotection in this sample?

- The decision to merge maternal and paternal PBI scores may obscure parent-specific effects; have the authors tested mother vs father bonding separately?

- The authors suggest patients with optimal attachment “learn effectively” pre-morbidly; can they cite developmental data supporting this protective mechanism?

- Clinical implications mention neuromodulation studies but do not connect directly to parenting interventions; I am curious about how realistic the proposal is to modify parental bonding retrospectively in adult populations.

- The affectionless control group was excluded due to imbalance—could an exploratory analysis retaining this category (with caution) be placed in Supplementary Material?

- Occasional grammatical slips (e.g., “was has been approved,” p.6) need correction.

**Do you want your identity to be public for this peer review?** For information about this choice, including consent withdrawal, please see our Privacy Policy

Reviewer #1: No

Reviewer #2: No

Reviewer #3: No

---

## [Author Response · Author response to Decision Letter 1]

19 Oct 2025

Response to Reviewer Comments

Journal Requirements:

Answer: The style guidelines have been implemented.

2. Please describe in your methods section how capacity to provide consent was determined for the participants in this study. Please also state whether your ethics committee or IRB approved this consent procedure. If you did not assess capacity to consent please briefly outline why this was not necessary in this case.

Answer: The capacity to consent was assessed and confirmed by the treating physicians for participating patients.

Answer: The information was added to the manuscript.

[This work was supported by the Forschungsförderungsfonds of the University Psychiatric Clinics of Basel. The authors declare financial support was received for the research.].

Answer: This work was financially supported by the “Forschungsförderungsfonds” of the University Psychiatric Clinics of Basel only. There were no other funds or sources of financial support involved. This information was added to the cover letter.

4. We note that Figure 1 includes an image of a patient/participant in the study.

Answer: The depiction of a human face in Figure 1 is a stimulus picture of the FER test and does not represent a participant in the study.The image of a human face in Figure 1 is a sample stimulus image as it was used in the Facial Emotion Recognition task A (FER A). The images are taken from the Karolinska Directed Emotional Faces picture set (KDEF; Lundquist, Flykt, & Öhman, 1998) with the full permission for use in research and publication. The ID of the specific image in Figure 1 is AF01SAS.

For the KDEF terms and conditions, see here:

https://kdef.se/faq/using-and-publishing-kdef-and-akdef

The origin of the stimulus images is stated in the description of the FER A in the Methods section of the manuscript. We noticed that the term used there, “Karolinska standard”, is not entirely correct, so we replaced it by the correct official name “Karolinska Directed Emotional Faces”.

Upon reviewing the KDEF terms and conditions, we also became aware that the KDEF publishers wish a list of all stimulus images used in the experimental paradigm to be published alongside the manuscript, so we added the full list of image IDs to the Supplementary Material and added a reference to the manuscript “A list of all stimulus image IDs used in the FER A paradigm can be found in the Supplementary Material”.

We further added the following statement to the caption of Figure 1: “The emotional face stimulus image is ID AF01SAS from the Karolinska Directed Emotional Faces set.

Answer: The citations have been updated.

Review Comments to the Author

Reviewer #1:

I have reviewed the manuscript titled “Facial Emotion Recognition Abilities of Individuals with Schizophrenia and the Influence of Parental Bonding – An Exploratory Study in a Forensic Sample.” The study addresses an important and underexplored area, investigating how parental bonding patterns influence facial emotion recognition (FER) performance in individuals with schizophrenia within a forensic sample. The findings are clinically relevant, as they suggest that optimal parental attachment may mitigate FER deficits, while neglectful parenting is associated with poorer performance. To further strengthen the study, I suggest the authors consider sharing their full analysis code and data processing pipeline (where ethically possible) to promote reproducibility.

Answer: The R packets used for the analysis and the description given in the manuscript should be sufficient for anybody seeking to reproduce the generalized linear model (GLM) as it was applied here. Nonetheless, we added the core analysis commands to the Supplementary Material.Study data in a forensic setting is very sensitive and protected more strictly by law and cannot be shared easily with the public.

-Including a more detailed subgroup or ablation analysis could help disentangle the specific contributions of different parental bonding dimensions (care vs. overprotection) to FER outcomes.

Answer: The sample size, respectively, the cell fillings resulting from the PBI classification is already at the limit of minimum statistical requirements and definitely too small to conduct subgroup analyses. The presented work is merely a secondary analysis of pre-existing data, however, a future study on parenting style and FER would be powered accordingly and allow for more sophisticated analyses. We are not familiar with the term “ablation analysis”.

-Validation on independent or more diverse cohorts would also help confirm the robustness of the findings. The discussion would benefit from a state of the art comparison table summarizing existing FER studies in schizophrenia to better contextualize the current results.

Deficits in facial emotion recognition (FER) are a well-established feature of schizophrenia, affecting both accuracy and response time. These deficits are present across different stages of the condition and in different risk groups. FER deficits in schizophrenia are consistent, particularly for negative emotions such as fear and anger, and are evident from the early stages of the condition and in individuals at high risk of developing it. These deficits are more pronounced and persistent in schizophrenia than in other psychotic disorders or in individuals without mental health conditions.

Answer: In addition to the studies already mentioned in the manuscript, we have summarized the current state of research on FER in schizophrenia in a table. A comparison of some recent studies of facial emotion recognition (FER) impairment across schizophrenia risk groups and control groups: See a Table in the file "Response to Reviewer Comments"

References

1. Bae, M., Cho, J., & Won, S. (2024). Facial emotion-recognition deficits in patients with schizophrenia and unaffected first-degree relatives. Frontiers in Psychiatry, 15. https://doi.org/10.3389/fpsyt.2024.1373288

2. Lee, S., Lin, G., Shih, C., Chen, K., Liu, C., Kuo, C., & Hsieh, C. (2021). Error patterns of facial emotion recognition in patients with schizophrenia.. Journal of affective disorders. https://doi.org/10.1016/j.jad.2021.12.130

3. Pena-Garijo, J., Lacruz, M., Masanet, M., Palop-Grau, A., Plaza, R., Hernández-Merino, A., Edo-Villamón, S., & Valllina, O. (2022). Specific facial emotion recognition deficits across the course of psychosis: A comparison of individuals with low-risk, high-risk, first-episode psychosis and multi-episode schizophrenia-spectrum disorders. Psychiatry Research, 320. https://doi.org/10.1016/j.psychres.2022.115029

4. Fusar-Poli, L., Pries, L., Os, J., Erzin, G., Delespaul, P., Kenis, G., Luykx, J., Lin, B., Richards, A., Akdede, B., Binbay, T., Altınyazar, V., Yalınçetin, B., Gümüş-Akay, G., Cihan, B., Soygür, H., Ulaş, H., Cankurtaran, E., Kaymak, S., Mihaljevic, M., Andrić-Petrović, S., Mirjanić, T., Bernardo, M., Mezquida, G., Amoretti, S., Bobes, J., Sáiz, P., García-Portilla, M., Sanjuán, J., Aguilar, E., Santos, J., Jiménez-López, E., Arrojo, M., Carracedo, Á., López, G., González-Peñas, J., Parellada, M., Maric, N., Atbaşoğlu, C., Üçok, A., Alptekin, K., Saka, M., Aguglia, E., Arango, C., O’Donovan, M., Rutten, B., & Guloksuz, S. (2021). Examining facial emotion recognition as an intermediate phenotype for psychosis: Findings from the EUGEI study. Progress in Neuro-Psychopharmacology and Biological Psychiatry, 113. https://doi.org/10.1016/j.pnpbp.2021.110440

5. Kuang, Q., Zhou, S., Liu, Y., Wu, H., Bi, T., She, S., & Zheng, Y. (2022). Prediction of Facial Emotion Recognition Ability in Patients With First-Episode Schizophrenia Using Amplitude of Low-Frequency Fluctuation-Based Support Vector Regression Model. Frontiers in Psychiatry, 13. https://doi.org/10.3389/fpsyt.2022.905246

6. Fusar-Poli, L., Pries, L., Van Os, J., Radhakrishnan, R., Pence, A., Erzin, G., Delespaul, P., Kenis, G., Luykx, J., Lin, B., Akdede, B., Binbay, T., Altınyazar, V., Yalınçetin, B., Gümüş-Akay, G., Cihan, B., Soygür, H., Ulaş, H., Cankurtaran, E., Kaymak, S., Mihaljevic, M., Andrić-Petrović, S., Mirjanić, T., Bernardo, M., Mezquida, G., Amoretti, S., Bobes, J., Sáiz, P., García-Portilla, M., Sanjuán, J., Aguilar, E., Santos, J., Jiménez-López, E., Arrojo, M., Carracedo, Á., López, G., González-Peñas, J., Parellada, M., Maric, N., Atbaşoğlu, C., Üçok, A., Alptekin, K., Saka, M., Aguglia, E., Arango, C., Rutten, B., & Guloksuz, S. (2022). The association between cannabis use and facial emotion recognition in schizophrenia, siblings, and healthy controls: Results from the EUGEI study. European Neuropsychopharmacology, 63, 47-59. https://doi.org/10.1016/j.euroneuro.2022.08.003

7. Hachtel, H., Deuring, G., Graf, M., & Vogel, T. (2024). Impact of psychosocial stress on facial emotion recognition in schizophrenia and controls: an experimental study in a forensic sample. Frontiers in Psychiatry, 15. https://doi.org/10.3389/fpsyt.2024.1358291

8. Kjellenberg, E., & Winblad, S. (2020). M74. FACIAL EMOTION RECOGNITION ABILITY IN PATIENTS WITH SCHIZOPHRENIA AND OTHER PSYCHOTIC DISORDERS. Schizophrenia Bulletin, 46, S163 - S163. https://doi.org/10.1093/schbul/sbaa030.386

9. Kuang, Q., Zhou, S., Li, H., Mi, L., Zheng, Y., & She, S. (2022). Association between fractional amplitude of low-frequency fluctuation (fALFF) and facial emotion recognition ability in first-episode schizophrenia patients: a fMRI study. Scientific Reports, 12. https://doi.org/10.1038/s41598-022-24258-7

10. Kuang, Q., Liu, Y., Zhou, S., Bi, T., Mi, L., She, S., & Zheng, Y. (2021). The correlation between fractional amplitude of low-frequency fluctuation-based resting-state functional magnetic resonance imaging and facial emotion recognition ability in patients with first-episode schizophrenia. **. https://doi.org/10.21203/rs.3.rs-919680/v1

- A dedicated limitations and future work section should be expanded to explicitly address issues such as sample size, gender restriction, retrospective bias in PBI reporting, and generalizability. Finally, the inclusion and exclusion criteria for participant selection should be clearly summarized for clarity.

Answer: In the section on strengths and weaknesses and discussion, the relevant information on sample size, gender restriction, retrospective bias in PBI reporting, and generalisability was added. The inclusion and exclusion criteria were outlined more clearly in the introduction.

- For further methodological inspiration, particularly on advanced feature extraction and classification in psychiatric populations, the authors may consider reviewing the following works: “Zipper Pattern: An Investigation into Psychotic Criminal Detection Using EEG Signals,” “Multilevel Hybrid Handcrafted Feature Extraction Based Depression Recognition Method Using Speech,” and “Automated Schizophrenia Detection Model Using Blood Sample Scattergram Images and Local Binary Pattern.” These studies illustrate how explainable and structured computational models can enhance the understanding and classification of complex psychiatric and neurocognitive phenomena offering valuable insights for future extensions of this important research on schizophrenia and emotion processing.

Answer: Thank you very much for this valuable information. We will gladly incorporate these methodological tips and insights into our future work.

Reviewer #2:

The article addresses a relatively unexplored issue: the role of parental attachment in facial emotion recognition (FER) among individuals with schizophrenia in forensic settings. Below some methodological and scientific limitations:

1. The sample size (n = 32) is limited and does not allow for generalizable conclusions. Additionally, one parenting style category (“affectionate constraint”) is not represented in the sample. Please, include a power analysis to justify the sample size or explicitly acknowledge the lack of such analysis as a limitation.

Answer: This manuscript presents an exploratory analysis of a secondary endpoint of the study published by Hachtel, H. et al. 2024 (MS reference 56), thus an a priori power analysis was conducted, but with respect to the primary outcome of that study and a different design.

(Hachtel H, Deuring G, Graf M, Vogel T. Impact of psychosocial stress on facial emotion recognition in schizophrenia and controls: an experimental study in a forensic sample. Front Psychiatry. 2024 Jul 16;15:1358291. doi: 10.3389/fpsyt.2024.1358291. PMID: 39081531; PMCID: PMC11287427.)

The commented power analysis itself can be downloaded here (open access): https://www.frontiersin.org/articles/10.3389/fpsyt.2024.1358291/full#supplementary-material

We additionally emphasized that secondary, exploratory analysis aspect in the manuscript.

It is inherently unpredictable how the four-categories PBI parenting style classification will turn out before a study, incidences are skewed among different populations. Our results may serve as a starting point for future studies on the subject.

The Generalized Linear Model (GLM) revealed medium to large effect sizes (Cohen's ωp² ≥ 0.06) and significant post hoc contrasts exhibited large effect sizes (Hedges’ g ≥ 1.34). When comparing the parameter and contrast estimates from the original analyses to those obtained via bootstrapping, only minimal discrepancies (≤ 3%) were observed, while bootstrapped post hoc contrast effect sizes (Hedges’ g) were estimated ~5% larger. This consistency between analytical and bootstrapped estimates supports our confidence in t

---

## [Decision Letter · Decision Letter 1]

10 Dec 2025

Facial emotion recognition abilities of individuals with schizophrenia and the influence of parental bonding  – an exploratory study in a forensic sample.

PONE-D-25-17462R1

Dear Dr. Salome Todua-Lennigk,

We’re pleased to inform you that your manuscript has been judged scientifically suitable for publication and will be formally accepted for publication once it meets all outstanding technical requirements.

Kind regards,

Mu-Hong Chen, M.D., Ph.D.

Academic Editor

PLOS One

Additional Editor Comments (optional):

Reviewers' comments:

Reviewer's Responses to Questions

**Comments to the Author**

Reviewer #3: All comments have been addressed

2. Is the manuscript technically sound, and do the data support the conclusions?

Reviewer #3: Yes

3. Has the statistical analysis been performed appropriately and rigorously?

Reviewer #3: Yes

4. Have the authors made all data underlying the findings in their manuscript fully available?

Reviewer #3: Yes

5. Is the manuscript presented in an intelligible fashion and written in standard English?

Reviewer #3: Yes

Reviewer #3: (No Response)

**Do you want your identity to be public for this peer review?** For information about this choice, including consent withdrawal, please see our Privacy Policy

Reviewer #3: No

---

## [Editor Report · Acceptance letter]

PONE-D-25-17462R1

PLOS One

Dear Dr. Todua-Lennigk,

I'm pleased to inform you that your manuscript has been deemed suitable for publication in PLOS One. Congratulations! Your manuscript is now being handed over to our production team.

Kind regards,

on behalf of

Dr. Mu-Hong Chen

Academic Editor

PLOS One